# Research on Fish Slicing Method Based on Simulated Annealing Algorithm

**Shuo Liu** [1,2], **Hao Wang** [2] **and Yong Cai** [3,*]

1   Key Laboratory of Ocean Observation-Imaging Testbed of Zhejiang Province, Zhejiang University, Zhoushan 316021, China; shuoliu@zju.edu.cn
2   Ocean Academy, Zhejiang University, Zhoushan 316021, China; haow97@zju.edu.cn
3   Ocean Research Center of Zhoushan, Zhejiang University, Zhoushan 316021, China
*   Correspondence: caiyong888@zju.edu.cn

**Abstract:** Multiobjective optimization is a common problem in the field of industrial cutting. In actual production settings, it is necessary to rely on the experience of skilled workers to achieve multiobjective collaborative optimization. The process of industrial intelligence is to perceive the parameters of a cut object through sensors and use machines instead of manual decision making. However, the traditional sequential algorithm cannot satisfy multiobjective optimization problems. This paper studies the multiobjective optimization problem of irregular objects in the field of aquatic product processing and uses the information guidance strategy to develop a simulated annealing algorithm to solve the problem according to the characteristics of the object itself. By optimizing the mutation strategy, the ability of the simulated annealing algorithm to jump out of the local optimal solution is improved. The project team developed an experimental prototype to verify the algorithm. The experimental results show that compared with the traditional sequential algorithm method, the simulated degradation algorithm designed in this paper effectively improves the quality of the target solution and greatly enhances the economic value of the product by addressing the multiobjective optimization problem of squid cutting. At the end of the article, the cutting error is analyzed.

**Keywords:** multiobjective optimization; simulated annealing; cutting optimization problem; cutting algorithm

## 1. Introduction

In the field of cut processing, workers usually rely on their own experience to cut products. Due to the different operating proficiencies of workers, many errors occur, which reduces the production efficiency of qualified products. With the continuous improvement in the industrial automation level, a variety of processing machinery and control algorithms have been designed to replace manual labor [1,2]. However, traditional mechanical algorithms cannot cut and optimize complex targets such as the human brain and cannot meet the ever-changing production requirements. For example, in the field of aquatic product processing, there are many single-objective optimization problems. Workers can use traditional machinery to perform fixed-weight cutting with the optimization goal of obtaining the same segment weights from a fish body [3–5]. For the goal of obtaining the same segment lengths, fixed-length segmentation of the fish body is performed. These studies have been widely used in the processing flow of factories. With the continuous improvements in production requirements, some multiobjective optimization problems have appeared in the field of aquatic product processing. Such problems cannot be solved by traditional machinery alone, and there are few related studies. The optimization problem of large squid slices to be solved in this paper is a multiobjective optimization problem.

The multiobjective optimization problem in this paper can be described as follows: After preprocessing a large squid (such as the giant squid D. gigas [6]), the processed raw materials in Figure 1a (length × width approximately 400 × 160 mm) are obtained.

According to the cutting plan shown in Figure 1b, the raw materials are cut to obtain the finished product in Figure 1c. After removing the first and last waste materials, each piece of the finished product is a small piece that achieves the target index. As shown in Figure 1d, the weight is $40 \pm 2$ g and the side diagonal connection length is $40 \pm 2$ mm. The diagonal connection is shown in Figure 1e. If the weight index or diagonal index of a small piece exceeds the allowable range, the small piece is judged as unqualified. The unqualified rate of the cut product will determine the ultimate economic benefits. Therefore, the raw material cutting problem is to determine the starting and ending cutting positions and angles for each small piece of raw material so that the above two goals can be optimized at the same time. Workers usually rely on their own experience and adopt different cutting strategies according to different squid shapes to cut raw materials to meet the above two parameters at the same time, but this process depends heavily on the experience of the workers. Even the most experienced workers have a failure rate of more than 20% for their cut products. To solve this problem, researchers have developed a device that can perform fixed weight cutting, but it cannot meet the diagonal size requirements at the same time. The research goal of this article is to further solve this multiobjective optimization problem on this basis.

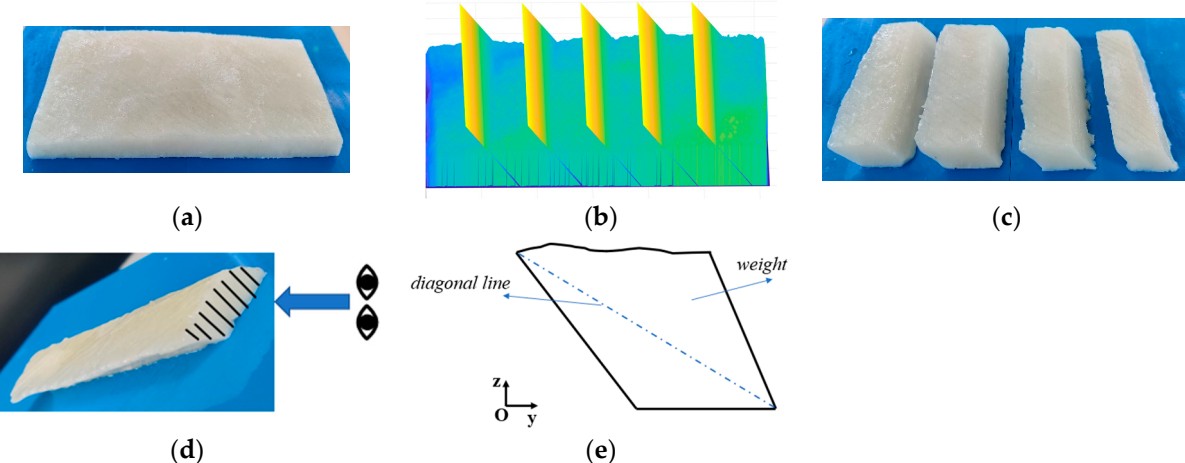

**Figure 1.** Squid image. (**a**) is the top view of the raw material; (**b**) is the raw material cutting plan; (**c**) is the product obtained from the b plan; (**d**) is a qualified product in (**c**); from the perspective of (**d**), (**e**) is the product simplified graphics.

In many industries (such as paper, cloth, metal plates, and wood), there are multi-objective cutting optimization problems, all of which aim to obtain small products that meet customer needs from large raw materials under the conditions of meeting multiple optimization objectives [7,8]. For example, in the textile industry, the fabric cutting position needs to be determined according to customer quality requirements. The flaws in the fabric itself will affect the cutting results. To minimize the impact of the flaws on the quality of the cloth and to make the continuous length of the cloth as large as possible, Ozdamar [9] used the simulated annealing algorithm to optimize these two goals. The convergence of the algorithm was sped up by maintaining an appropriate mutation rate. Compared with the sequential algorithm, the simulated annealing algorithm increases the length of the good-quality cloth by 75%, which greatly improves the economic value of the cloth. The optimization problems in the forestry, steel and textile industries are similar in nature, but the applications of the problems are different due to the characteristics of the cutting material, the quality grading, and the needs of customers. In many industries, the cut size and type of expected products are fixed, and it is necessary to optimize the arrangement and combination of cutting methods for raw materials of different sizes. However, changing the cutting mode will increase the cutting cost. The total benefit of cutting is determined by the number of cut products and the cost of changing the cutting mode. Yanasse and Limeira [10] used a hybrid heuristic algorithm to optimize the two

goals of the number of cut products and the cost of changing cutting modes. Golfeto [11] proposed combining a genetic algorithm and the biological symbiosis relationship to solve this problem; they used the interaction relationship among the optimization target parameters, combined multiple targets into a fitness function through the weighted sum method, and compared different weights. The experiments showed that the accuracy and time consumption of this method were within a reasonable range. Cui and Liu [12] proposed a sequential heuristic algorithm to first meet the condition of the largest number of finished products and then reduced the cost of changing cutting modes. He adjusted the calculation structure of the fitness function and reduced the total cost while reducing the calculation time by an order of magnitude. Mobasher [13] found that the hybrid linear programming algorithm has difficulty solving such problems, so he proposed two local search algorithms and a heuristic algorithm based on column generation for comparison. The results of the comparative test showed that the heuristic algorithm is better than other algorithms in calculating the cost of different modes. Araujo [14] used genetic algorithms to optimize the two objective functions for minimizing material waste and minimizing the number of cutting patterns. The fitness of the different targets in the function were sorted, and the smallest combination of blade positions was used to generate the largest number of target products and the least waste. Through an actual dataset test, it was found that the algorithm can shorten the calculation time from 60 to 80 min to approximately 5 s while maintaining a high accuracy rate. Ronnqvist [15] summarized the forestry industry's logging problem, logistics transportation and other optimization models. In summary, for the multiobjective cutting optimization problem, traditional methods are more difficult to solve. Researchers began to use intelligent algorithms to solve the cutting problem and mainly studied the initialization of the solution in intelligent algorithms, mutation problems, the definitions of objective functions and other issues. They adopted many strategies, such as real number solutions, binary code solutions, dictionary sort objective functions, and weighted objective functions. Research on these problems in terms of intelligent algorithms, the actual combination of intelligent algorithms and industrial problems has gradually become the mainstream research direction [16].

The optimization problem to be solved in this paper belongs to the field of aquatic product processing, and the raw materials for segmentation are very different from those in the abovementioned literature. First, different from the single-objective optimization problem of fixed-weight and fixed-length segmentation of fish bodies, this article explores a multiobjective optimization problem, which includes the optimization of two target parameters, weight and shape parameters, according to the irregular shape of the raw material. In addition, in the abovementioned studies, most of the objects to be cut, such as cloth, glass, and steel, were regular shapes, which can be described by mathematical functions. The object studied in this paper is a naturally grown fish body. Its shape is relatively complicated and cannot be described by an accurate mathematical model. It is necessary to use a laser scanning method to reconstruct the fish body in three dimensions and then complete the cut based on the scanned model. Therefore, higher requirements are put forward for the robustness of the algorithm. The above factors have increased the difficulty in solving the problem and make it difficult for traditional sequential algorithms to obtain the optimal solution of the problem while considering the multiple optimization objectives at the same time. Therefore, this paper developed a simulated annealing algorithm to solve the multiobjective cutting optimization problem of irregular fish-body shapes.

This article is divided into the following sections. Section 2 describes the problem to be solved and its mathematical model. Section 3 introduces the simulated annealing algorithm used in this article. Section 4 introduces the application results of the simulated annealing algorithm on the actual data set and analyzes the error. Conclusions and further research directions are given in Section 5.

## 2. Problem Description

To better explain the multiobjective optimization problem of fish body cutting mentioned in the first part, this section explains the three aspects of the fish body data source, target parameter calculation and objective function establishment.

### 2.1. Data Acquisition

The first step in solving this cutting problem is to obtain input data. After being processed and frozen, the squid body to be cut becomes smooth on the lower surface and uneven on the upper surface, as shown in Figure 2a. As shown in Figure 2b, the research group used the Gocator 2150 laser displacement sensor located directly above the raw material for scanning. After tilt correction and Gaussian filtering, the point cloud image in Figure 2c is obtained. After integration processing, the true three-dimensional shape of the fish body is obtained.

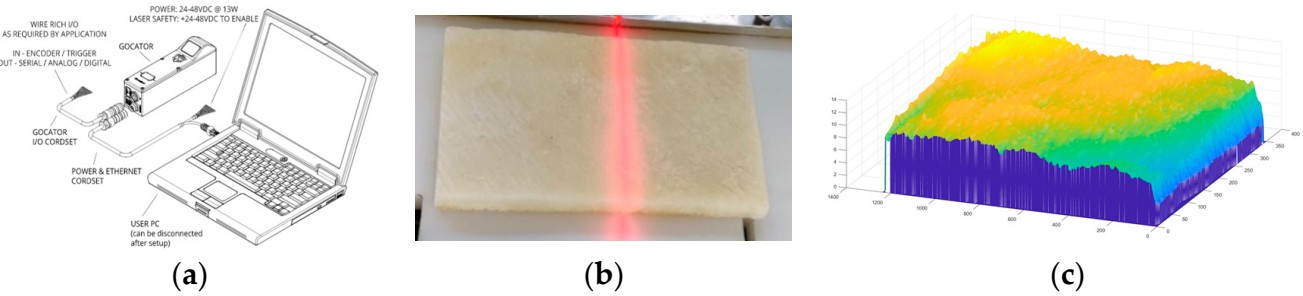

(a)　　　　　　　　　　　　　　　(b)　　　　　　　　　　　　　　　(c)

**Figure 2.** (**a**,**b**) Fish body scanning process and (**c**) fish body point cloud image.

### 2.2. Parameter Calculation

Next, the raw material in Figure 2a is divided into multiple small pieces (as shown in Figure 1d) and the score of each piece is calculated. To describe the problem conveniently, first, two cutting quality evaluation parameters, "weight" and "diagonal length", are defined. Weight refers to the physical weight of each small block and is determined by the volume and density of the small block. The diagonal length represents the length of the diagonal line of the diamond on the side projection surface of the small piece, as shown in Figure 1e. As mentioned above, the purpose of cutting is to obtain as many fish pieces as possible that weigh $40 \pm 2$ g and have a diagonal length of $40 \pm 2$ mm. When dividing, the oblique cutting method is adopted instead of the vertical cutting method. As shown in Figure 3, the angles between the front and rear cutting surfaces of each small piece and the horizontal plane are described as angle (1) and angle (2).

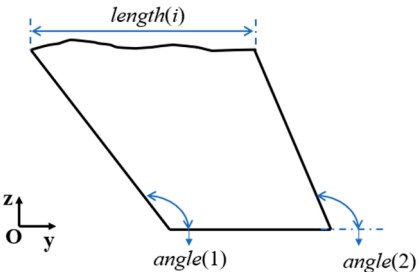

**Figure 3.** Schematic diagram of cutting parameters.

Assuming that the error between the actual weight and the expected weight of each piece is *errorW* and the error between the diagonal length and the expected length is *errorDL*, then the score is proportional to *errorW* and *errorDL*. The *errorW* and *errorDL* of each piece are determined by the actual angle and actual length of the piece. The ultimate goal of this cutting problem is to minimize the score in the cutting process.

To better describe the calculation method of the objective function value score, this article assumes that the angle and length of a piece are known and discusses the calculation methods of the target parameters *errorW* and *errorDL*.

### 2.2.1. Determination of *errorW*

The first target parameter, *errorW*, is determined by the difference between the actual weight of the piece (*realW*) and the expected weight (*idealW*). For a fish body with uniform density $\rho$, it is determined by the difference between the actual volume of the block (*realVol*) and the expected volume (*idealVol*) is as follows:

$$
\begin{aligned}
errorW \; &= (realW - idealW)/idealW \\
&= (\rho \cdot realVol - \rho \cdot idealVol)/(\rho \cdot idealVol) \\
&= (realVol - idealVol)/idealVol
\end{aligned} \tag{1}
$$

Therefore, to determine *errorW*, first the value of *realVol* is determined. The calculation method of *realVol* is as follows.

According to the point cloud image shown in Figure 2, the smallest volume element in mm can be obtained. The volume element is considered to be the smallest unit of a 3D object with dimensions of $w$, $l$, and $h$.

The top image information after threshold processing is used as the input of the model, and the area enclosed by the top image and the horizontal zero point is the effective area of the fish body section. Assuming that the entire raw material can be divided into $N$ sections (Figure 4 shows the point cloud image of a certain section) and the volume element number of each section is $Nv_i$, $i = 1 \ldots N$, the area of each section (unit: mm$^2$) is as follows.

$$
A_i = N_{vi} \cdot w \cdot h, \; i = 1 \ldots N \tag{2}
$$

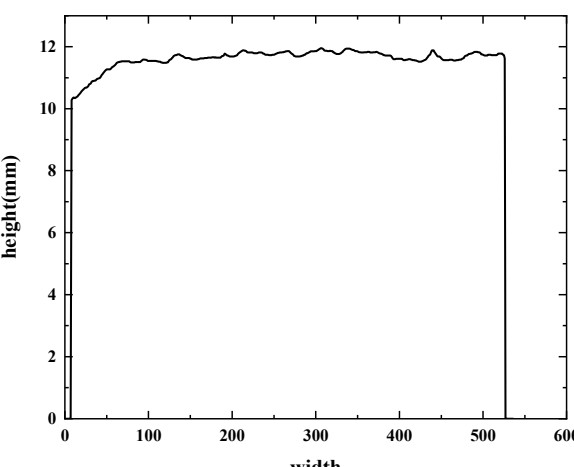

**Figure 4.** Example of a scanned cross-sectional image of a fish body.

The volume between two consecutive sections (unit: mm$^3$) is calculated as follows:

$$
V_i = A_i \cdot l, \; i = 1 \ldots N \tag{3}
$$

However, the cutting problem in this article does not consider vertical guillotine cutting [17]; therefore, the cutting angle needs to be considered. The long fish body shown in Figure 1a can be simplified into a rhombus-like body along the viewing direction, as shown in Figure 5.

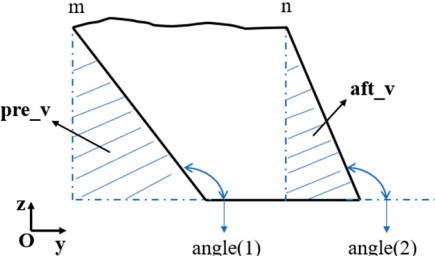

**Figure 5.** The front and rear angles cause changes in the volume calculation method.

Due to the existence of the front and rear cutting angles, angle (1) and angle (2), the actual volume needs to be subtracted from the front hollow volume *pre_v* after adding *aft_v*:

$$realVol = \sum_{i=m...n} V_i + aft\_v - pre\_v \tag{4}$$

### 2.2.2. Determination of *errorDL*

The second target parameter *errorDL* is determined by the product customization requirements of aquatic processing plants.

The *realW* of each small piece can be directly determined by the volume, and for the diagonal length error errorDL, to address the appearance characteristics of the product, it is necessary to comprehensively judge the diagonal lengths *leftDL* and *rightDL* on the left and right sides. The specific calculation methods of *leftDL* and *rightDL* are shown in Figure 6.

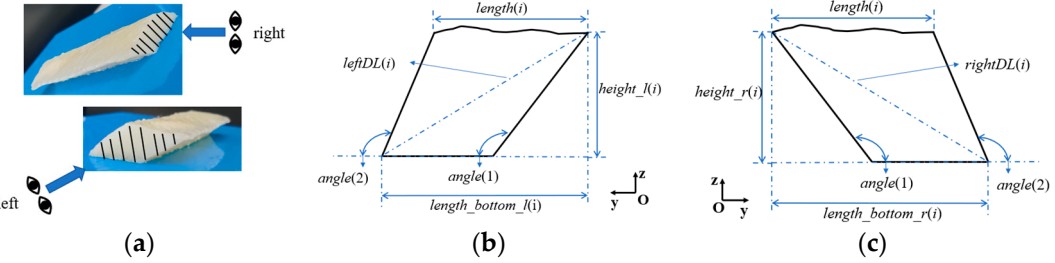

| (**a**) | (**b**) | (**c**) |

**Figure 6.** Diagonal length calculation method. Looking at the side of the fish from the left and right directions in (**a**), you can obtain the simplified image on the left side (**b**) and the simplified image on the right side (**c**).

Among them, the calculation method of *rightDL* is as follows:

$$rightDL = \sqrt{height\_r^2 + length\_bottom\_r^2} \tag{5}$$

where *height_r* represents the height value of the section and *length_bottom_r* represents the length of the bottom surface. The calculation method of *leftDL* is similar to that of *rightDL*. Then, *errorDL* can be described as follows:

$$errorDL = \frac{(leftDL + rightDL)/2 - idealDL}{idealDL} \tag{6}$$

where *idealDL* is the length of the target diagonal and is a certain constant.

### 2.3. Object Function

The goal of the calculation model is to minimize the *score* as much as possible. In the above section, the calculation of the parameters *errorW* and *errorDL* is explained, and then, how to determine the *score* based on the above parameters is described. Table 1 describes the main parameters in the calculation process. This article takes a certain cutting plan as an example to illustrate the specific calculation process.

**Table 1.** Main parameters used for the cutting problem.

| Symbol | Description |
|:---:|:---:|
| $w, l, h$ | volume element unit (length, width, height) |
| *idealVol* | expected volume of small piece |
| *idealDL* | expected diagonal length of small piece |
| $n$ | maximum number of pieces of the whole raw material |
| *length(i)* | The length of the *i*-th piece |
| *angle(i), angle(i+1)* | front and back cutting angle of the *i*-th piece |
| *realVol(i)* | the actual cutting volume of the *i*-th piece |
| *rightDL(i)* | the length of the right diagonal of the *i*-th piece |
| *leftDL(i)* | the left diagonal length of the *i*-th piece |
| *score(i)* | the score of the *i*-th piece |
| *Score* | the sum of the scores of all pieces |

The cutting plan is shown in Figure 7. In this scheme, there are *n* = 4 small pieces. Combining Formulas (1), (4), (5) and (6), the score value of each piece can be obtained:

$$score(i) = w1 \cdot |errorW(i)| + w2 \cdot |errorDL(i)|, \ i = 1 \ldots n \tag{7}$$

where *w*1 and *w*2 represent the weights of *errorW* and *errorDL*, respectively, in the score. According to the factory's emphasis on quality and shape, *w*1 = *w*2 = 0.5.

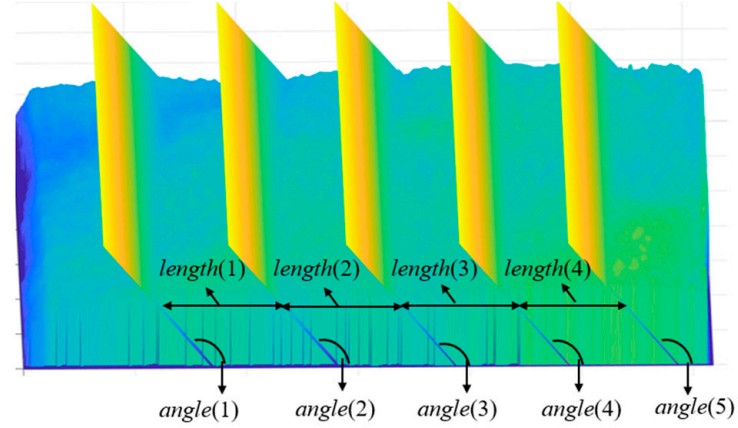

**Figure 7.** Schematic diagram of fish body cutting plan.

Then, the cutting quality score of the entire raw material is:

$$Score = \sum_{i=1}^{n} score(i) \tag{8}$$

In summary, this article introduces a calculation method for the target parameters *errorW* and *errorDL* and a method for composing the target function value score. Next, we consider how to use optimization algorithms to reduce *errorW* and *errorDL* to make the quality score reach an ideal value and then solve the multiobjective optimization problem proposed in this article.

## 3. SA algorithm for the Cutting Problem

After determining the calculation method for the quality score, this paper proposes an intelligent algorithm to reduce the value of the score. This article describes the algorithm from the following three aspects: the choice of the intelligent algorithm, the components and the optimization of the algorithm.

### 3.1. Selection of an Intelligent Algorithm

Algorithm selection is the first step in algorithm design. The basic principle of optimization algorithm selection is that the algorithm must be able to fit the characteristics of the problem, and the most important impact on the performance of the intelligent optimization algorithm is the distribution characteristics of the solution. Therefore, the optimization algorithm can be selected by determining the difference of the distribution density function [18].

This article randomly selects a piece of raw material fish data. The raw material can be divided into 10 small pieces, with the *score* of each small piece being ≤0.05 as the standard; the ideal range of the *Score* is ≤0.5. We divide the feasible solution into samples by uniformly sampling 10,000 times. The value range of the objective function is divided into a series of cells, and the frequency of the value interval of the objective function is used for distribution statistics. The statistical results are shown in Figure 8.

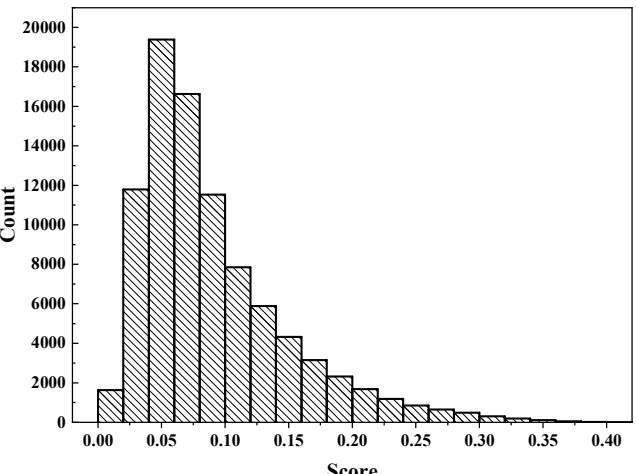

**Figure 8.** Frequency distribution of target value.

In Figure 8, at both ends of the value range, the probability of the optimal solution drops rapidly. According to the statistical results of the knowledge base in [14], for this type of distribution density problem, the simulated annealing (SA) algorithm has a higher solution accuracy and can accelerate the convergence of the model to the optimized solution. Therefore, to solve the precise fish body cutting problem, this paper developed a more effective simulated annealing algorithm.

The basic principle of the SA algorithm is to optimize the parameters on the basis of the initial solution to find a final solution that satisfies the termination condition or the value of the objective function. Due to the irregular shape of the fish body and other factors, there are a large number of local solutions to the problem. This is a major challenge for the algorithm's optimization ability, and it is necessary to jump out of the local solution in time to approach the global optimum. Therefore, this paper uses the information guidance strategy to improve the simulated annealing algorithm.

### 3.2. Cutting Algorithm

After determining the use of the SA algorithm, this section will explain the basic principles of the SA algorithm and the optimizations made for the problems in this article. After determining the initial temperature T, it mainly includes four steps: the determination of the initial solution, the generation of a new solution, the Metropolis criterion, and the cooling criterion. The calculation steps of the SA algorithm are as follows (Algorithm 1):

---
**Algorithm 1**: SA algorithm

---
**Input:** number of iterations iter; initial temperature T; current solution; inner loop
**Output:** best solution
    1.    iter=0
    2.    initialise T
    3.    stop criterion = maximum number of iterations
    4.    initialise current solution
    5.    current cost = Evaluate(current solution)
    6.    **while** not stop criterion **do**
    7.        **while** inner loop **do**
    8.            Neighbour = Generate(current solution)
    9.            Neighbour cost = Evaluate(Neighbour)
    10.            **if** Accept(current cost, Neighbour cost, T)
    11.                current solution = Neighbour
    12.                Current cost = Neighbour cost
    13.            **end**
    14.            Update(best solution, iter)
    15.        **end**
    16.        Update(T)
    17.        Update(stop criterion)
    18.    **end**
    19.    **return** best solution

---

- Initial solution
- Since the cuts must be continuously distributed throughout the fish body, the starting position of the initial solution must be determined, which is determined by the algorithm's preprocessing strategy. After determining the starting position, use the real number vector to establish the initial solution, the size of which is $2n + 1$, that is, $X = [x1, x2, \ldots, x2n + 1]$. The first $n$ elements are the length of each small piece, and the $n + 1$th to $2n+1$th elements are the cutting angles of the front and back sides of each small piece, so $X$ can also be expressed as $[length(1) \ldots length(n), angle(1) \ldots angle(n+1)]$.
- Generation of new solutions

In the standard intelligent algorithm, the pure mutation operation is not instructive, and the efficiency is low, and there are many local solutions to this cutting problem, and it is easy to fall into the local optimum. If the knowledge accumulated in the search process can be combined, it will help improve the search performance of the algorithm.

This paper uses the information-guided simulated annealing algorithm [19], and uses the change trend of the solution in two adjacent iterations as the next search direction for the individual. For example, for the optimization problem min f(x), the population size is $N$, and the $k$-th generation individuals are $X1 (k)$, $X2 (k)$, ..., $XN (k)$. Introduce a vector $D_i(k) = (d_i^1, d_i^2, \ldots, d_i^n)^T, i = 1, 2, \ldots, N$ to record the next search direction of the individual $X_i(k) = (x_i^1(k), x_i^2(k), \ldots, x_i^n(k))^T$, where $d_i^j = sign(x_i^j(k) - x_i^j(k-1))$. Let $G(i)$ denote the algebra of individual i's survival. According to rule I, record the survival algebra $G$ of the individual and the search direction vector $D$ of the next step, and update it at each step.

Rule I: If $f(X_i(k)) < f(X_i(k-1))$,

Then $G(i)=1$, $D_i(j) = (sign(x_i^j(k) - x_i^j(k-1)))^T$;
Otherwise, $G(i) = G(i-1) + 1$, where $i = 1, 2, \ldots, N; j = 1, 2, \ldots, n$.
Rule II: If $G(i)=1$,

Then $\sigma_i(k) = \exp(-k/\alpha)$, $X_i(k)' = X_i(k) + D_j(k) \cdot \left| N(0, \sigma_i(k)) \right|$;

Otherwise, move one step randomly. The moving step length is related to the survival algebra of the individual and the value of the function,

$$\sigma_i(k) = \exp(-k/\alpha) \cdot G(i), \quad X_i(k)' = X_i(k) + \left| N(0, \sigma_i(k)) \right|$$

According to Rule I and Rule II, if the performance of the current solution is better than that of the previous generation and random perturbation is performed while maintaining the search direction to obtain a new solution; if the performance of the current solution is worse than the previous generation, random disturbance obtains a new solution; if the search falls into a certain local optimal solution, then as the individual survival algebra continues to increase, the amplitude of the disturbance also increases, helping to deviate from the local optimal solution.

The algorithm will produce infeasible solutions in the process, so it is necessary to check the feasibility of the newly generated solution and make adjustments. The method used in this paper is: if the variable exceeds the feasible range, based on the boundary value, make a new solution $x' = boundary + |x' - boundary|$.

- Metropolis Guidelines

In the cutting stage, after calculation, if the fitness function of the solution is $f(S)$, the fitness of the current solution ret1 is $f(S1)$, and the fitness of the new solution ret2 generated according to the current solution ret1 is $f(S2)$. According to Metropolis criterion, if $df = f(S2) - f(S1) < 0$, it means that the new solution ret2 is better than the current solution ret1. Replace ret1 with ret2, otherwise accept the new solution with probability of $\exp(-df/T)$.

The Metropolis guideline is:

$$p = \begin{cases} 1, df < 0 \\ \exp(-\frac{df}{T}), df \geq 0 \end{cases} \tag{9}$$

- Cool down

Use the cooling rate q for cooling, that is $T = qT$. In each cycle, if $T$ is less than the end temperature, stop the iteration and output the current state, otherwise continue the iteration.

This section describes the calculation process of the simulated annealing algorithm and the corresponding optimization strategy. Next, consider combining it with actual data to solve the multi-objective optimization problem of this article, and analyze the effect of the algorithm.

## 4. Results and Discussions

In this section, the effect of the simulated annealing algorithm in solving the actual cutting problem is verified, and it is analyzed and compared with the sequential algorithm. Then, the source of the error is analyzed.

### 4.1. Implementation on Real Data

This research is based on experimental prototypes and has been tested in a factory. The dataset contains several pieces of raw material randomly collected from the factory. Due to the natural growth of fish, the length and height of each piece of raw material are different, and the number of pieces $N$ ranges from 8 to 14 pieces. To test the effect of the simulated annealing algorithm, this paper compares the results with those of the sequential algorithm (Figure 9 shows the processing flow chart).

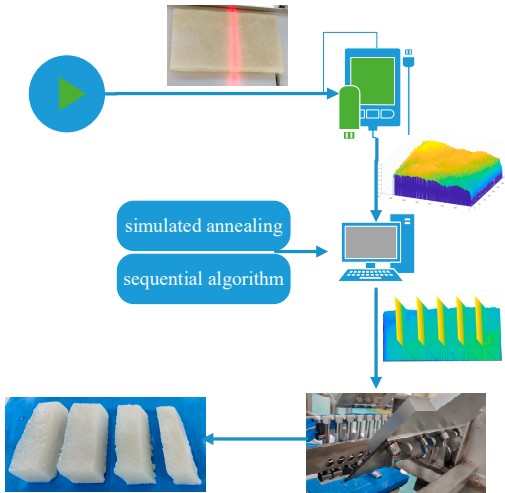

**Figure 9.** Processing flow chart.

### 4.1.1. Sequential Algorithm

The goal of the sequential algorithm is to make the weight of each piece reach the expected standard. Starting with the first subsection, the algorithm extends the first piece millimeter by millimeter until the current size (which implies the ending location of the first piece) violates the requirements of the highest weight category. At that point, the algorithm decreases the size by one millimeter and checks the requirement for the corresponding category. If the constraint is satisfied, then the process ends at the current piece size; otherwise, the current size is increased by one millimeter, and the piece's weight is compared with the corresponding category requirement. This process is repeated for each piece. Figure 10 describes the calculation steps of the sequential algorithm.

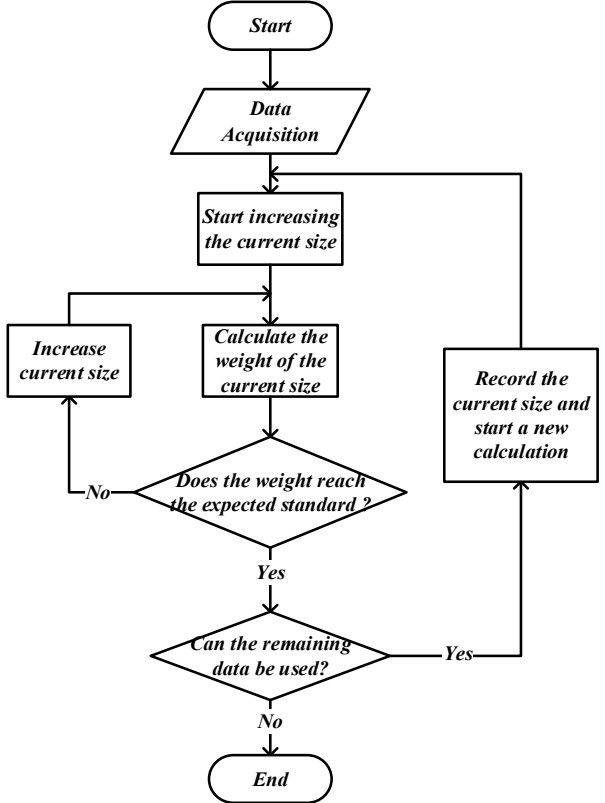

**Figure 10.** The calculation steps of the sequential algorithm.

### 4.1.2. Results of the Two Algorithms

The raw materials obtained in the factory can be divided into 50 pieces. The calculation standard for each piece is *idealW* = 40 g, *diagIdeal* = 40 mm, $-5\% \leq$ errorW <= 5%, $-5\% \leq$ *errorDL* $\leq$ 5%, and *score* $\leq$ 5%. When the simulated annealing algorithm is applied, the maximum number of iterations is 4000; when more than 4000 iterations are used, the algorithm starts multiple times, and no improvement is found. Two algorithms were used to perform independent calculations, and the results were recorded for analysis.

Table 2 describes the *errorW* and *errorDL* values of 50 small blocks calculated by the two algorithms and visually compares the algorithm results with those in Figures 11 and 12. Table 3 describes the statistical results of *errorW*; they are denoted as follows: the maximum (*maxW*), minimum (*minW*), average (*avgW*), standard deviation (*stdW*), and ratio greater than $-5\%$ and less than 5% (*rateW*); the statistical results of *errorDL* are denoted as follows: the maximum (*maxDL*), minimum (*minDL*), mean (*avgDL*), standard deviation (*stdDL*), ratio greater than $-5\%$ and less than 5% (*rateDL*).

**Table 2.** Results of the experiment.

| | Sequential Algorithm | | | | | | Simulated Annealing | | | | |
| n | P1 [1] | P2 [2] | n | P1 | P2 | n | P1 | P2 | n | P1 | P2 |
|---|---|---|---|---|---|---|---|---|---|---|---|
| 1 | 0 | −0.048 | 26 | 0 | 0.044 | 1 | −0.008 | −0.056 | 26 | 0.007 | −0.05 |
| 2 | 0 | −0.157 | 27 | 0 | 0.038 | 2 | −0.007 | −0.018 | 27 | −0.001 | −0.019 |
| 3 | 0 | −0.174 | 28 | 0 | 0.045 | 3 | −0.002 | −0.013 | 28 | 0.008 | −0.067 |
| 4 | 0 | −0.185 | 29 | 0 | 0.067 | 4 | 0.007 | −0.005 | 29 | 0.002 | −0.003 |
| 5 | 0 | 0.032 | 30 | 0 | −0.141 | 5 | −0.016 | −0.012 | 30 | 0.006 | −0.041 |
| 6 | 0 | 0.034 | 31 | 0 | −0.159 | 6 | 0.021 | −0.012 | 31 | 0.011 | −0.023 |
| 7 | 0 | 0.043 | 32 | 0 | −0.181 | 7 | 0.009 | 0.007 | 32 | −0.009 | −0.035 |
| 8 | 0 | 0.052 | 33 | 0 | −0.18 | 8 | 0.013 | −0.009 | 33 | 0.025 | −0.051 |
| 9 | 0 | 0.076 | 34 | 0 | −0.023 | 9 | −0.003 | −0.011 | 34 | 0.003 | −0.012 |
| 10 | 0 | −0.042 | 35 | 0 | −0.011 | 10 | 0.005 | −0.008 | 35 | 0.006 | −0.019 |
| 11 | 0 | −0.148 | 36 | 0 | −0.018 | 11 | −0.004 | −0.05 | 36 | −0.005 | −0.011 |
| 12 | 0 | −0.17 | 37 | 0 | −0.019 | 12 | 0.001 | −0.043 | 37 | 0.005 | −0.008 |
| 13 | 0 | −0.168 | 38 | 0 | −0.018 | 13 | 0 | −0.039 | 38 | 0.016 | 0.028 |
| 14 | 0 | −0.065 | 39 | 0 | −0.017 | 14 | 0.003 | −0.051 | 39 | 0.024 | 0.004 |
| 15 | 0 | −0.074 | 40 | 0 | 0.019 | 15 | 0.005 | −0.046 | 40 | −0.022 | −0.016 |
| 16 | 0 | −0.013 | 41 | 0 | −0.02 | 16 | 0.003 | −0.027 | 41 | −0.011 | −0.06 |
| 17 | 0 | −0.009 | 42 | 0 | −0.187 | 17 | −0.003 | −0.047 | 42 | 0.04 | −0.07 |
| 18 | 0 | −0.08 | 43 | 0 | −0.193 | 18 | 0.002 | −0.012 | 43 | 0.009 | −0.001 |
| 19 | 0 | −0.048 | 44 | 0 | −0.192 | 19 | 0.009 | −0.058 | 44 | −0.016 | −0.017 |
| 20 | 0 | −0.172 | 45 | 0 | 0.062 | 20 | −0.001 | −0.047 | 45 | 0.015 | −0.024 |
| 21 | 0 | −0.044 | 46 | 0 | 0.062 | 21 | 0.006 | −0.037 | 46 | −0.002 | −0.052 |
| 22 | 0 | −0.178 | 47 | 0 | 0.049 | 22 | 0 | −0.041 | 47 | 0.005 | 0.002 |
| 23 | 0 | 0.046 | 48 | 0 | 0.058 | 23 | 0.004 | −0.046 | 48 | 0.003 | −0.001 |
| 24 | 0 | 0.043 | 49 | 0 | 0.064 | 24 | 0.008 | −0.039 | 49 | 0.023 | 0 |
| 25 | 0 | 0.046 | 50 | 0 | 0.075 | 25 | −0.002 | −0.034 | 50 | 0.053 | −0.003 |

[1] P1 stands for *errorW*, [2] P2 stands for *errorDL*.

**Table 3.** Result statistics.

| Parameters | Statistics | Sequential Algorithm | Simulated Annealing |
|---|---|---|---|
| *errorW* | *maxW* | 0% | 5.31% |
| | *minW* | 0% | −2.16% |
| | *avgW* | 0% | 0.49% |
| | *stdW* | 0% | 1.3% |
| | *rateW* | 100% | 98% |
| *errorDL* | *maxDL* | 7.59% | 2.83% |
| | *minDL* | −19.32% | −6.96% |
| | *avgDL* | −4.36% | −2.61% |
| | *stdDL* | 9.4% | 2.25% |
| | *rateDL* | 48% | 90% |

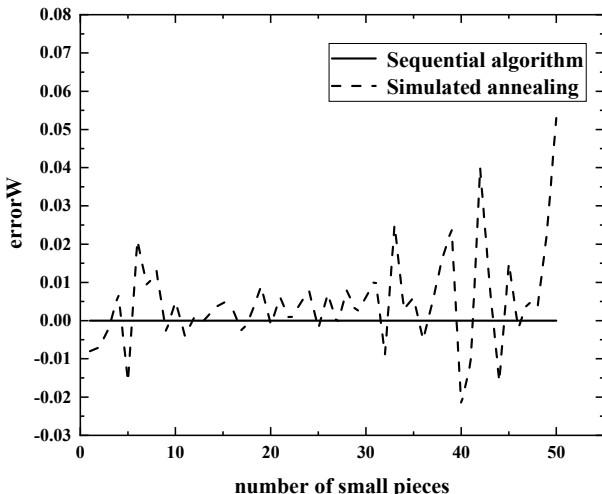

**Figure 11.** Comparison of errorW.

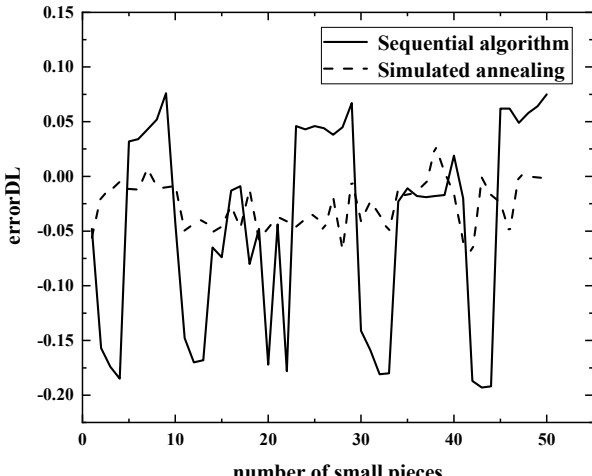

**Figure 12.** Comparison of errorDL.

As seen from the above table and image, for 50 pieces, the *errorW* calculated by the sequential algorithm is 0%, so the pass rate in terms of volume error is 100%. Some of the SA calculation results are greater than 5%, and the pass rate is 98%, which is slightly lower than that of the sequential algorithm. In terms of the *errorDL* results, the numerical value of the sequential algorithm varies widely, and the overall pass rate is 48%. The SA is controlled within a good range. As shown in Figure 12, the result is better optimized than that of the sequential algorithm, and the pass rate is increased to 90%.

Considering Formula (7), the score of each small piece is the weighted sum of *errorW* and *errorDL*. Considering that the factory attaches the same importance to the weight and shape, the errors of both must be guaranteed to meet the requirements, so $w1 = 0.5$, $w2 = 0.5$. Then, the score distribution obtained by using the two algorithms is shown in Figure 13. Table 4 shows the score obtained by the two algorithms; the maximum value is denoted as *maxSc*, the minimum value is denoted as *minSc*, the average value is denoted as *avgSc*, the standard deviation is denoted as *stdSc*, and the ratio of the scores that are less than 5% is denoted as *rateSc*.

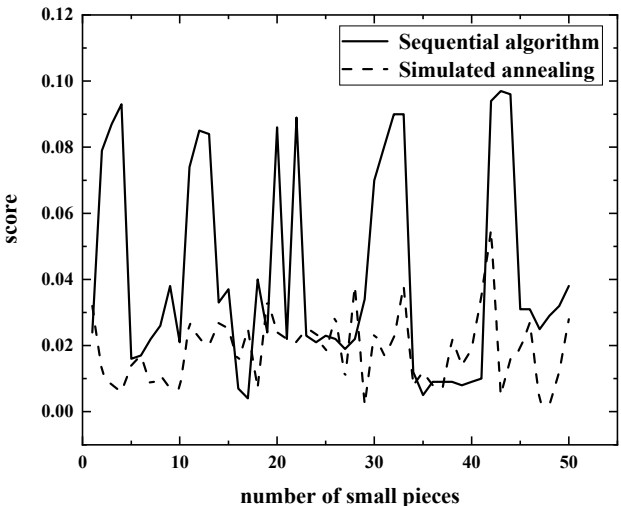

**Figure 13.** Comparison of score.

**Table 4.** Score result statistics.

|  | *maxSc* | *minSc* | *avgSc* | *stdSc* | *rateSc* |
|---|---|---|---|---|---|
| Sequential algorithm | 9.66% | 0.44% | 4.09% | 3.14% | 70% |
| Simulated annealing | 5.49% | 0.22% | 1.86% | 1.09% | 96% |

According to Figure 13 and Table 4, compared to the sequential algorithm, the simulated annealing algorithm has greatly improved the results and does very well in optimizing the score. The overall pass rate increased from 70% to 96%. Therefore, the simulated annealing algorithm developed in this paper is suitable for solving the squid multiobjective cutting optimization problem.

### 4.2. Error Analysis

According to the above results, the simulated annealing algorithm developed in this paper is superior to the sequential algorithm, and the overall satisfaction rate of the *score* can reach 96%. However, in terms of the results of the two target parameters *errorW* and *errorDL*, the simulated annealing algorithm does not have a higher optimization effect for either parameter. There are slight errors between the parameter values of some small pieces and the expected value. This section analyzes these errors.

First, $|errorW|$, $|errorDL|$, and the quality score are placed in the same image, as shown in Figure 14. In the figure, the value of 2.5% is used as the dividing line, and the data can be divided into four categories: A: $|errorW| \leq 2.5\%$ and $|errorDL| \leq 2.5\%$; B: $|errorW| \leq 2.5\%$ and $|errorDL| > 2.5\%$; C: $|errorW| > 2.5\%$ and $|errorDL| \leq 2.5\%$; and D: $|errorW| > 2.5\%$ and $|errorDL| > 2.5\%$. The percentage of each type of data is shown in Table 5.

**Table 5.** Proportion of four types of data.

|  | **A** | **B** | **C** | **D** |
|---|---|---|---|---|
| percent | 50% | 46% | 2% | 2% |

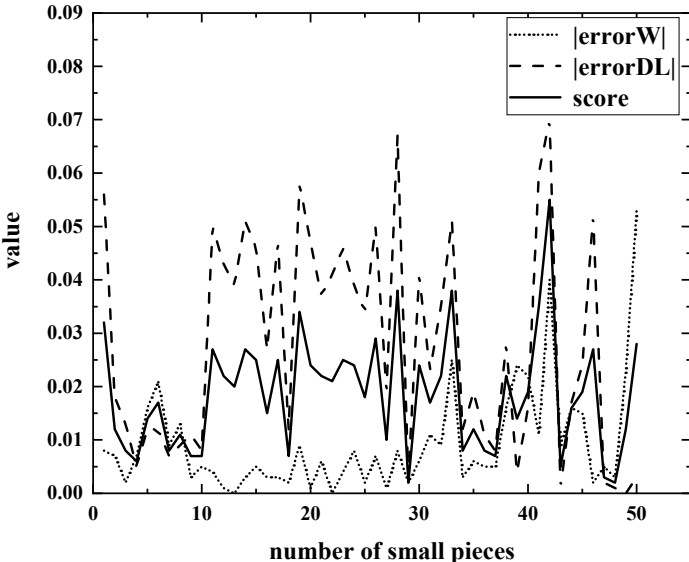

**Figure 14.** Comparison of $|errorW|$, $|errorDL|$, score.

From the above data, it can be seen that for 50% of the small pieces, the error is maintained within a good range. However, 46% of small blocks still have large $|errorDL|$ errors. This is caused by the physical properties of the raw materials. The raw material of a naturally grown fish body has an uneven height on the upper surface. The following model is considered for calculating the volume and diagonal:

$$\begin{aligned} volume &= length \cdot width \cdot height \\ diagonal &= \sqrt{length^2 + height^2} \end{aligned} \qquad (10)$$

Assuming that the width is fixed, the uneven height of the upper surface of the raw material will cause a significant change in the height. When the height is large, to obtain the expected volume value, the length will need to be reduced, which will cause a large error in the diagonal value diagonal. When the height is small, to obtain the expected volume value, the length will need to be increased, which will also cause diagonal errors. For the other 4% of small blocks, the causes of the errors are similar.

Therefore, the optimization result obtained by the SA algorithm may be located at the edge of the optimal set. The irregularity of the natural body of a fish limits the improvement in the optimization results.

## 5. Conclusions

This paper studies the application of the improved simulated annealing algorithm to the multiobjective optimization and cutting problem of irregular fish in the aquatic field, uses the information guidance strategy to optimize the generation of the target solution, and completes the optimization of the two target parameters of fish body weight and shape.

Through the comparison experiment with the sequential algorithm, it can be seen that the simulated annealing algorithm effectively improves the quality of the target solution and increases the pass rate of the target parameter *errorDL* by 42% and the pass rate of the quality score by 26%; hence, it has a very significant improvement effect. Due to the irregularity of the target object, the accuracy of the target solution is limited. However, the algorithm can still complete the solution task well. This method greatly improves the economic value of the product, is conducive to the standardization of product specifications, greatly reduces the labor intensity of workers and has broad application prospects in the field of aquatic product processing. Therefore, the simulated annealing algorithm developed in this paper is suitable for solving the abovementioned multiobjective problem.

**Author Contributions:** Conceptualization, S.L. and H.W.; methodology, S.L. and H.W.; validation, H.W.; resources, S.L. and Y.C.; formal analysis, S.L. and H.W.; investigation, S.L., H.W. and Y.C.; writing, S.L. and H.W.; funding acquisition, S.L. and Y.C. All authors have read and agreed to the published version of the manuscript.

**Funding:** This work was supported in part by the National Key R&D Program of China under Grant 2019YFD0901603 and the Zhejiang Province Key R&D Program under Grant 2019C04017.

**Institutional Review Board Statement:** Not applicable.

**Informed Consent Statement:** Not applicable.

**Data Availability Statement:** Not applicable.

**Acknowledgments:** The authors are most grateful to all subjects for their participation in the study. The author would like to thank the aquatic product processing factory which provided the experimental raw materials and the help of other researchers.

**Conflicts of Interest:** The authors declare no conflict of interest.

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
