# Peer review of "Research on Fish Slicing Method Based on Simulated Annealing Algorithm"

_applsci, doi:10.3390/app11146503_

Round 1
Reviewer 1 Report
Multi objective optimization is a common problem in the field of industrial cutting. It is necessary to rely on the experience of skilled workers to achieve multi objective collaborative optimization. The process of industrial intelligence is to perceive the parameters of a cut object through sensors and use machines instead of manual decision making. However, the traditional sequential algorithm cannot satisfy multi objective optimization problems. This paper studies the multi objective optimization problem of irregular objects in the field of aquatic product processing and uses the information guidance strategy to develop a simulated annealing algorithm to solve the problem according to the characteristics of the object itself. It uses the information guidance strategy to optimize the generation of the target solution, and completes the optimization of the two target parameters of fish body weight and shape. Through the comparison experiment with the sequential algorithm, it can be seen that the simulated annealing algorithm effectively improves the quality of the target solution and increases the pass rate of the target parameter error by 42% and the pass rate of the quality score by 26%; hence, thus it has a significant improvement effect. Due to the irregularity of the target object, the accuracy of the target solution is limited. However, the algorithm can still complete the solution task well. Hence, this method improves the economic value of the product, is conducive to the standardization of product specifications, reduces the labor intensity of workers and has broad application prospects in the field of aquatic product processing. I recommend publishing the paper after minor revision concerning its introduction Section which should be improved, especially with adequate literature. I also congratulate authors for a job well done.
Reviewer 2 Report
some simple mistakes are founded:
- in line 49: (c) must change to Figure 1.c
- in line 151: (c) must change to Figure 2.
Description of SA algorithm is given in Fig. 9, but I don't found description of sequential algorithm in 4.1.1.
What are the authors' initials for references 14 and 15.
The classical SA algorithm predicts the achievement of "thermal equilibrium". Has this been taken into account?
It is not really clear how it is inferred from the Figure 12 that "... pass rate is increased to 90% ".
Experimental part is fulfilled.
